# The impact of farmers' digital literacy on farmers' enthusiasm for grain cultivation: Findings from the "Double Hundred and Thousand" farmers in Jiangxi province

Zhipeng Wang, Yuqin Song, Huiping Xie, Chaoqun Li, Yujie Hu, Zhihua Wu*, Jianghua Chen¤*

School of Economics and Management, Jiangxi Agricultural University, Nanchang, China

¤Current address: Jiangxi Agricultural University, Nanchang City 330045, Jiangxi, China
* wuzhihua0831@126.com (ZW); jianghuachen@jxau.edu.cn (JC)

## Abstract

Enhancing farmers' enthusiasm for grain cultivation is crucial for ensuring national food security. Based on survey data from the "Double Hundred and Double Thousand" farmers in Jiangxi Province, this study measures farmers' enthusiasm for grain cultivation through their double-cropping rice cultivation behavior and employs Probit and Tobit models to empirically analyze the impact of farmers' digital literacy on their enthusiasm for grain cultivation in traditional double-cropping rice production areas of Jiangxi, as well as its underlying mechanisms. The findings reveal: (1) Improving farmers' digital literacy exerts a significantly positive influence on their enthusiasm for grain cultivation, promoting double-cropping rice adoption and expanding cultivation areas in traditional production regions. This conclusion remains robust after rigorous testing. (2) Mechanism analysis identifies three pathways: accelerating farmland transfer-in, facilitating the utilization of agricultural machinery socialization services, and promoting the adoption of green grain production technologies, through which enhanced digital literacy drives double-cropping rice cultivation and elevates cultivation enthusiasm. (3) Heterogeneity analysis demonstrates that the positive effect of digital literacy on double-cropping rice cultivation is more pronounced among farmers with higher educational attainment and the level of household part-time farming. Therefore, policies should prioritize elevating farmers' overall digital literacy, narrowing digital capability gaps, refining land transfer procedures and mechanisms, leveraging the advantages of farmers' cooperatives, and advancing the dissemination of agricultural machinery socialization services and green grain production technologies.

**Data availability statement:** All relevant data are within the paper and its Supporting Information files.

**Funding:** This work was supported by the National Natural Science Foundation of China Youth Project (Award Number: 42201294 | Recipient: Zhipeng Wang), the National Natural Science Foundation of China Regional Project (Award Number: 72263018 | Recipient: Jianghua Chen), Humanities and Social Sciences Research Fund of the Ministry of Education(Award Number: 21YJC790127 | Recipient: Zhihua Wu), the Jiangxi Provincial Social Science Foundation Project (Award Number: 22YJ34 | Recipient: Zhipeng Wang), the China Postdoctoral Surface Fund Project (Award Number: 2023M731433 | Recipient: Zhipeng Wang), the Jiangxi Provincial University Humanities and Social Science Research Project (Award Number: JC22248 | Recipient: Zhipeng Wang) and the Research on Integrated Prevention and Control Pathways for Citrus Huanglongbing at County Level in Jiangxi Province: A Management Science Project (2025)(Award Number: 20252BAA100020 | Recipient: Zhipeng Wang).

**Competing interests:** The authors have declared that no competing interests exist.

# 1 Introduction

"Food is the foremost governance priority, and grain security anchors national stability". Grain security, vital to national economic and social development, constitutes a critical foundation for safeguarding national security. Under the scientific guidance and practical innovations of seminal theories such as the "Bowl Theory", "Red Line Theory", and "Grain Bottom Line Theory", China achieved a historic "20th consecutive bumper harvest" in 2023. However, the national grain security system still faces multiple challenges, with farmland abandonment emerging as a prominent threat to stable production [1,2]. In traditional double-cropping rice production areas, the shift from double-cropping to single-cropping rice has intensified seasonal abandonment, significantly reducing land multiple cropping indices and negatively impacting grain output. For instance, Jiangxi Province, a traditional double-cropping rice region, experienced declines of 12.3% and 19.6% in early and late rice sowing areas in 2022 compared to 2015, while mid-season and single-cropping late rice areas surged by 134.1%, reflecting waning farmer enthusiasm for double-cropping rice [3]. To address these challenges, central and local governments have implemented policies to bolster double-cropping rice production. At the central level, since 2023, the national government has allocated 2.475 billion yuan to support centralized seedling cultivation infrastructure in southern regions. Initiatives like the Grain Yield Enhancement and Efficiency Innovation Project promote balanced rice yield increases, with dedicated funding from the Major Grain-Producing Counties Incentive Program targeting double-cropping rice. At the local level, governments have also introduced supportive measures. For example, Hunan Province integrated multiple agricultural funds in 2023—including surplus farmland fertility subsidies, crop rotation funds, and commodity grain province rewards—to incentivize double-cropping rice. These policies aim to enhance farmer motivation, stabilize and expand cultivation areas, and strengthen grain production capacity.

Farmers' enthusiasm for grain cultivation is a critical factor in safeguarding national food security, and thus has garnered significant attention from scholars globally. Research in this domain primarily explores the following perspectives: First, from the national food security angle, Altieri et al. emphasized that sustaining farmers' cultivation motivation is a prerequisite for food security, arguing that only by protecting farmers' grain production profits and ensuring profit maximization can national food security be guaranteed [4]. Second, from agricultural policy perspectives, enhancing effective agricultural supply and increasing farmers' income are vital for boosting cultivation enthusiasm, aligning with China's current agricultural economic and policy objectives. However, debates persist regarding the impact of grain subsidy policies. From the perspective of farmers' income, Yi et al. concluded that agricultural subsidies will increase the total income and agricultural income of farmers [5], which is conducive to improving cultivation enthusiasm. However, many scholars have found that agricultural subsidies have limited incentive effect on farmers' grain production [6,7]. Third, from farmer income perspectives, agricultural insurance can play the role of risk dispersion, loss compensation and protection of farmers' interests, which is conducive to the improvement of grain production scale and enthusiasm for grain

cultivation [8]. Fourth, from farmer production behavior perspectives, Hu et al. demonstrated that farmers' enthusiasm for growing grain is related to traffic convenience, natural environment and family affection factor [9]. As micro-level agents in grain production, farmers' enthusiasm directly determines the stable growth of grain output and the realization of national food security strategic goals. Continued exploration of factors influencing cultivation enthusiasm holds significant theoretical and practical value for refining food security systems.

With the advent of the digital economy era, the deep integration of agricultural production and digital governance in China has amplified the empowering role of digital technologies in farming, positioning digital literacy as a critical nexus for farmers to engage with modern agriculture [10], and becoming a new variable influencing grain planting decisions. Farmers' digital literacy encompasses multifaceted capabilities, including accessing information through digital networks, interpreting information, and applying it to solve practical problems, spanning dimensions such as digital learning, social, and transaction literacy [11]. Existing studies demonstrate that enhancing farmers' digital literacy facilitates timely and precise access to market information, reduces production costs, expands social networks, and accumulates social capital, thereby promoting rural entrepreneurship [12–14], strengthening ecological cognition [15,16], and boosting participation in and responsiveness to digital financial initiatives [17]. Moreover, the existing research has achieved certain results in the field of farmers' grain production behavior and digital literacy effect, and it is of certain value to further explore the influence mechanism of farmers' digital literacy on farmers' enthusiasm for grain cultivation.

Building on this, in order to improve farmers' enthusiasm for grain cultivation and promote agricultural production and income increase, this study takes the traditional double-cropping rice production area of Jiangxi Province as the research area, and deeply explores the influence mechanism and action path of farmers' digital literacy on farmers' enthusiasm for grain cultivation. The following contributions are made:

First, the evaluation system of farmers' digital literacy suitable for rural China is constructed from five dimensions: digital media literacy, digital information literacy, digital social literacy, digital financial literacy and digital technology literacy. Second, it identifies farmland transfer-in, agricultural machinery socialization services, and adoption of green grain production technologies as intermediary mechanisms. Third, the heterogeneity analysis is carried out according to educational attainment and the level of household part-time farming. Fourth, grounded in theoretical and empirical findings, targeted recommendations are proposed to elevate farmers' digital literacy and amplify cultivation motivation, delivering actionable strategies to enhance agricultural income and safeguard national food security.

## 2 Theoretical analysis and research hypotheses

### 2.1 Impact of farmers' digital literacy on enthusiasm for grain cultivation

According to farmer behavior theory, farmers, as "rational economic agents", pursue profit maximization. Thus, the enthusiasm for grain cultivation among farmers in traditional double-cropping rice production areas is influenced by the economic value derived from their decisions [18]. Concurrently, information search theory posits that information and cost constraints are primary barriers to farmer decision-making. As a critical component of human capital, digital literacy directly affects farmers' production and management practices [19]. Digital literacy enables farmers to leverage digital technologies for real-time, efficient, and low-cost resource integration, information dissemination, and communication. This informational advantage reduces transaction costs, increases grain cultivation income, and thereby enhances enthusiasm for grain cultivation in traditional double-cropping rice regions.

On the one hand, farmers can utilize digital tools (e.g., websites, agricultural service apps) to precisely search for, acquire, and transmit high-value agricultural information. Enhanced capabilities in processing and analyzing information allow them to make informed production decisions, lowering information search and acquisition costs. Digital platforms also expand production and sales channels, mitigating information asymmetry in agricultural product markets, reducing transaction intermediaries, and strengthening bargaining power to lower negotiation costs. On the other hand, in production, the application of digital technologies such as the Internet of Things, drones, and artificial intelligence in

double-cropping rice cultivation to accurately control all aspects of agricultural production, which can optimize the allocation efficiency of agricultural capital, labor, and land, thereby elevating economic returns [20]. In sales, participation in e-commerce enables farmers to secure higher reservation prices, boost sales volumes, and expand cultivation profits.

Based on this, we propose Hypothesis 1:

H1: Enhancing farmers' digital literacy positively influences their enthusiasm for grain cultivation.

## 2.2 Mechanism analysis of farmers' digital literacy on enthusiasm for grain cultivation

### 2.2.1 Farmland transfer-in enhances cultivation enthusiasm.
Farmers' digital literacy can accelerate farmland transfer-in, thereby increasing their enthusiasm for grain cultivation and incentivizing double-cropping rice production. Farmers with high digital literacy typically leverage digital platforms to proactively access emerging trends in agriculture, rural areas, and farmer-related domains, breaking information barriers and translating external informational advantages into agricultural management capabilities and improving production efficiency. This effectively promotes agricultural production scaling, elevates demand for farmland transfer-in, and expands operational scale. Meanwhile, expanded operational scale necessitates two critical adjustments: Increased Labor Commitment: Households must allocate more primary labor to agriculture, heightening their income dependence on agricultural activities, which serve as the main channel for income growth [21]. Capital Inflow: Digital finance provides a pathway for high-digital-literacy farmers to reduce information asymmetry and lower transaction costs in land transfers, diversify funding sources, and alleviate financial constraints during land transfer processes [22], thereby securing financial stability for agricultural production. Additionally, driven by government policies such as "Prime Farmland for Grain Use" and agricultural subsidies, high-digital-literacy farmers, as "rational economic agents", exhibit stronger preferences for double-cropping rice cultivation to enhance land utilization rates and achieve yield and income growth.

Based on this, we propose Hypothesis 2:

H2: Farmers' digital literacy enhances enthusiasm for grain cultivation by accelerating farmland transfer-in.

### 2.2.2 Utilization of agricultural machinery socialization services enhances farmers' enthusiasm for grain cultivation.
Farmers' digital literacy promotes the use of agricultural machinery socialization services, thereby increasing their enthusiasm for grain cultivation and incentivizing double-cropping rice production. On the one hand, urbanization, rural labor migration, farmland transfer, and expanded agricultural operational scales create objective conditions for adopting machinery socialization services. On the other hand, as societal digitalization advances, farmers' digital literacy improves. Specifically, high-digital-literacy farmers leverage internet platforms and new media to broaden agricultural information channels, and the increase of agricultural information channels is conducive to the use of agricultural machinery by farmers and stimulates farmers' utilization of machinery services [23].

Agricultural machinery socialization services enhance cultivation enthusiasm through three mechanisms: Specialized Labor Division: Optimizing resource allocation, reducing labor inputs, alleviating labor shortages during "double-cropping rush seasons", improving production efficiency, and increasing yields and incomes. Cost Reduction: Addressing rising production costs and financial constraints on machinery purchases, these services overcome resource endowment limitations, reconfigure existing resources, and lower production and transaction costs [24]. Risk Mitigation: During "double-cropping rush seasons", machinery services enable rapid harvesting of early rice when threatened by typhoons or other disasters, minimizing potential losses and reducing risks [21]. Therefore, the adoption of agricultural machinery socialization services enhances farmers' enthusiasm for grain cultivation and promotes double-cropping rice production.

Based on this, we propose Hypothesis 3:

H3: Farmers' digital literacy enhances enthusiasm for grain cultivation by promoting the utilization of agricultural machinery socialization services.

### 2.2.3 Adoption of green grain production technologies enhances farmers' enthusiasm for grain cultivation.
Farmers' digital literacy facilitates the adoption of green grain production technologies, thereby increasing

their enthusiasm for grain cultivation and incentivizing double-cropping rice production. Farmers with high digital literacy tend to utilize digital technologies to address practical agricultural production challenges. Existing studies indicate that enhanced digital literacy promotes the adoption of green technologies through pathways such as enhancing farmers' risk perception, expanding farmers' digital social capital, and strengthening the effectiveness of technology promotion [25].

The adoption of green production technologies enhances cultivation enthusiasm through two mechanisms: Risk Mitigation and Cost Reduction: Technologies such as green pest control and information-based management strengthen resilience to agricultural risks, while reducing fertilizer and pesticide use lowers production costs, thereby boosting enthusiasm. Yield and Income Improvement: These technologies alleviate the problem of soil fertility degradation, increase production and income, and improve farmers' agricultural productivity and enthusiasm for grain cultivation [26]. In the long term, green practices (e.g., recycling of agricultural film and pesticide packaging, integrated water-fertilizer systems, and eco-efficient cultivation) reduce non-point source pollution, enrich soil fertility, and stabilize production outputs. Additionally, farmers adopting these technologies qualify for subsidies (e.g., green manure incentives), further increasing income. To maximize profits, farmers are more inclined to cultivate double-cropping rice.

Based on this, we propose Hypothesis 4:

H4: Farmers' digital literacy enhances enthusiasm for grain cultivation by promoting the adoption of green grain production technologies.

### 2.3 Heterogeneity analysis of farmers' digital literacy on enthusiasm for grain cultivation

Farmers constitute a highly heterogeneous group [27], and the impact of digital literacy on cultivation enthusiasm may vary across those with different endowment characteristics. First, farmers with varying educational levels exhibit significant disparities in their comprehensive abilities to acquire, comprehend, apply, and critically evaluate digital information, as well as their receptiveness to digital technologies. Those with higher education possess greater digital literacy, enabling them to leverage internet platforms and new media to acquire agricultural knowledge, scientifically and rationally manage cultivation, and prefer double-cropping rice production to expand cultivation areas for higher income, thereby enhancing enthusiasm. Second, the level of household part-time farming reflects the differences in resource allocation between the agricultural and non-agricultural sectors, which influences their technology adoption and production decisions [28]. Highly digitally literate farmers are more inclined to optimize agricultural production through market-oriented approaches, such as using digital platforms to access real-time market information and price signals to adjust crop structures and sales strategies; leveraging online channels to purchase agricultural inputs and obtain remote agricultural technical services, thereby reducing production costs and enhancing expected returns. This efficiency-improvement mechanism driven by digital literacy plays a more pronounced role among farmers with a higher level of household part-time farming, as they place greater emphasis on the marginal returns and resource allocation efficiency of agricultural operations. Consequently, digital literacy more effectively incentivizes double-cropping rice cultivation and enhances enthusiasm for grain cultivation.

Based on this, we propose Hypothesis 5:

H5: Farmers' digital literacy more effectively enhances cultivation enthusiasm among those with higher educational attainment and the level of household part-time farming.

Based on the above analysis, the theoretical analysis framework of the impact of farmers' digital literacy on farmers' enthusiasm for grain cultivation and the path is constructed, as shown in Fig 1.

## 3 Data source, variable setting and model selection

### 3.1 Data source

The study adopted a cross-sectional observational design using structured survey data derived from the "Double Hundred and Double Thousand" farmer survey conducted in 2023 by Jiangxi Agricultural University. The survey covered 216 villages across 72 townships in 24 counties (districts) of 11 prefecture-level cities in Jiangxi Province. Jiangxi Province

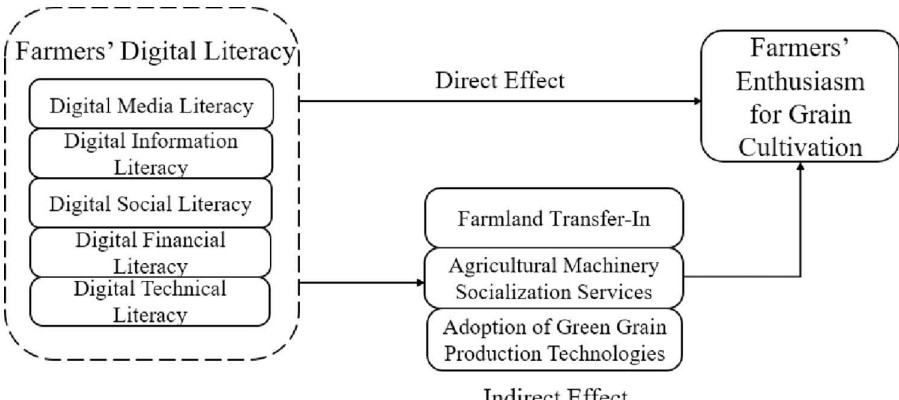

**Fig 1. The mechanism of the effect of farmers' digital literacy on farmers' enthusiasm for grain cultivation.**

was selected as the research area for the following reasons: First, it possesses a geographical and climatic environment suitable for double-cropping rice cultivation, with a long-standing history as a demonstration province for the "Rice Green, High-Quality, and High-Efficiency Initiative". Second, Jiangxi is a core province in China's double-cropping rice production regions, ranking among the top in national rice cultivation area. In 2024, the province's early rice sowing area reached 18.102 million mu (1.21 million hectares), accounting for 25.4% of the national total and ranking second nationwide.

The survey employed a stratified random sampling method. Counties were categorized into high-, medium-, and low-economic development tiers. Eight counties were systematically selected from each tier, followed by three townships per county based on nighttime light data. Three villages were surveyed per township, with 10 households per village, focusing on individual characteristics, household attributes, village features, and agricultural production practices. The fieldwork covered 2,160 households. To align with the research objectives, data filtering was conducted as follows: First, households cultivating rice were identified using the "input-output" section of the survey. Second, relevant questionnaire items were selected based on research needs and literature. Finally, incomplete, extreme, or erroneous responses were excluded, resulting in 921 valid questionnaires.

### 3.2 Variable setting

**3.2.1 Dependent variable.** The dependent variable in this study is farmers' enthusiasm for grain cultivation, measured through behavioral proxies such as double-cropping rice cultivation decisions and cultivation area. Compared to single-cropping rice, double-cropping rice requires greater agricultural resource inputs and larger cultivation areas, reflecting higher cultivation enthusiasm. Specifically: "Whether to cultivate double-cropping rice" is a binary variable, coded as 1 for adopters and 0 for non-adopters. "Double-cropping rice cultivation area" quantifies the scale of adoption as a continuous variable.

As shown in Table 2, 60.7% of sampled farmers cultivate double-cropping rice, indicating potential for further improvement in farmland utilization efficiency. Notably, the raw cultivation area data exhibit significant variability and scale discrepancies with other variables. To address this, the cultivation area variable is log-transformed for analysis.

**3.2.2 Core explanatory variable.** The core explanatory variable in this study is farmers' digital literacy. Drawing on existing research on digital literacy [29–31], we measure farmers' digital literacy across five dimensions—digital media, information, social, financial, and technical literacy—subdivided into 14 items. To ensure objectivity and accuracy in weighting, the entropy method is applied to assign weights to these indicators, thereby calculating the farmers' digital literacy index. Specific indicators, variable definitions, and assignments are detailed in Table 1.

**Table 1. Measurement system of farmers' digital literacy.**

| Dimension | Measurement | Items coding | Attribute |
|---|---|---|---|
| Digital media literacy | Ownership of smartphones | Yes = 1; No = 0 | + |
| | Broadband internet access | Yes = 1; No = 0 | + |
| | Ability to independently download mobile apps | Yes = 1; No = 0 | + |
| Digital information literacy | Use of internet to search for agricultural information | Yes = 1; No = 0 | + |
| | Ability to verify authenticity of online information | Yes = 1; No = 0 | + |
| | Preference for solving problems via online queries | Yes = 1; No = 0 | + |
| Digital social literacy | Sharing agricultural information via WeChat | Yes = 1; No = 0 | + |
| | Participation in village public affairs through social platforms | Yes = 1; No = 0 | + |
| | Expressing opinions via mobile internet | Yes = 1; No = 0 | + |
| | Creating/posting short videos using mobile apps | Yes = 1; No = 0 | + |
| Digital financial literacy | Usage of third-party payment platforms (e.g., WeChat Pay, Alipay) | Yes = 1; No = 0 | + |
| | Engagement with internet wealth management (e.g., Yu' e Bao, Ant Credit Pay, JD Digits credit products) | Yes = 1; No = 0 | + |
| | Online procurement of agricultural inputs | Yes = 1; No = 0 | + |
| Digital technical literacy | Adoption of digital technologies (e.g., IoT, UAVs, AI) in production | Yes = 1; No = 0 | + |

**3.2.3 Control variables.** Drawing on existing studies, this research controls for other potential factors influencing farmers' enthusiasm for grain cultivation by selecting control variables from individual characteristics, household characteristics, and village characteristics to mitigate empirical biases.

For individual characteristics, key variables include gender, age, educational attainment, health status, and satisfaction with grain purchase prices. Gender differences influence household agricultural decisions, as male household heads may exhibit greater proactivity in adopting digital technologies, thereby affecting cultivation behaviors. Aging farmers often experience reduced labor capacity, leading to shifts toward single-cropping rice or reduced double-cropping rice areas, thereby diminishing cultivation enthusiasm. Higher-educated farmers more easily access digital resources to optimize planting decisions and enhance cultivation income, boosting enthusiasm. Health status affects labor supply capacity, with most surveyed farmers reporting good health. Satisfaction with grain purchase prices also shapes enthusiasm, as farmers are more inclined to cultivate when satisfied with pricing.

For household characteristics, variables include the number of agricultural laborers, social capital, cultivated land area, paddy field fertility, the level of household part-time farming, land transfer behavior, and agricultural income share. The number of agricultural laborers directly determines household labor capacity, with labor-abundant households better equipped for double-cropping rice. Social capital enhances resource acquisition, as farmers with strong social networks leverage mutual aid to elevate enthusiasm. Cultivated land area is selected based on individual circumstances to optimize benefits and machinery utilization, thereby increasing enthusiasm. Paddy field fertility directly impacts yields, with farmers owning fertile land more inclined toward double-cropping rice. Farmers with a high level of household part-time farming are more likely to utilize digital technology as a tool for agricultural management from a market-oriented perspective, thereby significantly boosting their enthusiasm for grain cultivation. Land transfer behavior reflects land resource allocation, with farmers acquiring additional land showing greater enthusiasm and preference for double-cropping rice. Agricultural income share reflects dependency on farming, as households with higher shares prioritize grain production and double-cropping rice for income growth.

For village characteristics, variables include village economic development level and distance to the county seat. These factors influence local agricultural infrastructure and non-farm employment opportunities, thereby shaping cultivation decisions.

**3.2.4 Mechanism variables.** Drawing on existing studies [21,32], this study selects farmland transfer-in, agricultural machinery socialization services, and green grain production technology adoption as mechanism variables to explore the specific pathways through which farmers' digital literacy influences their enthusiasm for grain cultivation. For agricultural machinery socialization services, adoption is measured as a non-negative integer (0–6) based on participation in six key rice production stages: seedling cultivation, land preparation, transplanting, fertilization, pesticide application, and harvesting. As shown in Table 2, farmers adopt an average of 1.9 machinery services, indicating relatively low utilization levels. Green grain production technologies are measured as a non-negative integer (0–7) encompassing integrated water-fertilizer technology, green pest control, eco-efficient cultivation, straw recycling, agricultural film and pesticide packaging recycling, water-saving irrigation (e.g., drip/sprinkler systems), deep tillage fertilization, and straw crushing for soil incorporation. Table 2 reveals that farmers adopt an average of 1.1 green technologies, suggesting limited application of these practices.

Definitions and descriptive statistics for all variables are systematically presented in Table 2.

## 3.3 Model selection

The dependent variable is farmers' enthusiasm for grain cultivation, where the double-cropping rice cultivation decision is a binary variable (0/1). As the Probit model is suitable for binary dependent variables and following the approach of Zhu M and Yang R [33], the Probit model is selected for empirical analysis. Accordingly, the following model is constructed:

**Table 2. Variable definitions and descriptive statistics.**

| Variable type | Variable name | Definition and coding | Mean | Std. Dev. |
|---|---|---|---|---|
| Dependent variables | Double-cropping rice cultivation decision | 0 = No; 1 = Yes | 0.607 | 0.489 |
| | Double-cropping rice cultivation area (log) | Sown area (mu), logarithmic transformation | 1.626 | 1.817 |
| Core explanatory variable | Farmers' digital literacy | Index calculated via entropy method | 0.324 | 0.231 |
| Control variables | Gender | 0 = Female; 1 = Male | 0.857 | 0.351 |
| | Age | Actual age (years) | 57.879 | 10.645 |
| | Educational attainment | 1 = No formal education; 2 = Primary school; 3 = Junior high; 4 = Senior high; 5 = College+ | 2.863 | 0.915 |
| | Health status | 1 = Very poor; 2 = Poor; 3 = Fair; 4 = Good; 5 = Excellent | 3.847 | 0.999 |
| | Satisfaction with grain purchase prices | 1 = Very dissatisfied; 2 = Dissatisfied; 3 = Neutral; 4 = Satisfied; 5 = Very satisfied | 1.953 | 0.857 |
| | Number of agricultural laborers | Number of family members engaged in farming | 1.658 | 0.952 |
| | Social capital | Social interaction expenditure (Log) | 7.690 | 2.517 |
| | Cultivated land area | Total farmland area (mu) | 47.302 | 170.705 |
| | Paddy field fertility | 1 = Very poor; 2 = Poor; 3 = Fair; 4 = Good; 5 = Excellent | 3.358 | 0.833 |
| | Cooperative membership | 0 = No; 1 = Yes | 0.228 | 0.420 |
| | Land transfer behavior | 0 = No; 1 = Yes | 0.558 | 0.497 |
| | Agricultural income share | Percentage of household income from farming (%) | 37.303 | 36.352 |
| | Village economic development level | 1 = Very low; 2 = Low; 3 = Medium; 4 = High; 5 = Very high | 3.309 | 0.727 |
| | Distance to county seat | Kilometers from village to county center | 31.266 | 22.828 |
| Mechanism Variables | Farmland transfer-in | Transferred-in farmland area (mu), logarithmic transformation | 0.565 | 0.822 |
| | Agricultural machinery socialization services | Number of mechanized services adopted (0–6) | 1.889 | 1.464 |
| | Adoption of green grain production technologies | Number of sustainable practices adopted (0–7) | 1.099 | 1.320 |

$$\Pr o(decision_i = 1) = \Phi(\alpha_1 digital_i + \beta_1 X + \mu_{1i}) \tag{1}$$

Where $digital_i$ is the core explanatory variable, X represents control variables (including individual characteristics, household characteristics, and village characteristics), $decision_i$ is the dependent variable (1 = adoption of double-cropping rice; 0 = non-adoption), and $\mu_{1i}$ denotes the random error term.

For the double-cropping rice cultivation area, the Probit model is no longer applicable. Since only cultivation area data for adopters are observable (non-adopters' values are truncated to zero), a Tobit model is constructed for regression:

$$Ln(area_i + 1) = \alpha_2 digital_i + \beta_2 X + \mu_{2i} \tag{2}$$

Where the dependent variable $area_i$ is the logarithm of the double-cropping rice cultivation area, and other variables align with Equation (1).

## 4 Empirical analysis

### 4.1 Baseline regression results

This study employs Stata 18.0 for estimation, with regression results passing significance tests. As shown in Model 2 of Table 3, farmers' digital literacy exerts a significantly positive effect on double-cropping rice cultivation decisions at the 1% statistical level, while Model 4 in Table 3 indicates that digital literacy significantly and positively influences cultivation areas at the 1% level, confirming that digital literacy enhances farmers' enthusiasm for grain cultivation, thereby validating Hypothesis H1. On the one hand, improved digital literacy increases farmers' information awareness, enabling more efficient access to market and policy insights on double-cropping rice via digital platforms. On the other hand, heightened digital literacy facilitates the adoption of new technologies and methods, encourages engagement with professional agricultural guidance, and strengthens experiential learning. These informational and technical advantages empower farmers to optimize cultivation structures, enhance efficiency, and bolster confidence in double-cropping rice decisions. Additionally, enhanced digital literacy improves farmers' capacity for market risk management, mitigating risks associated with expanding cultivation areas, which collectively drives the increase in double-cropping rice planting scale.

From the perspective of control variables, the impact of cultivated land area on farmers' enthusiasm for grain cultivation is significantly positive, which may be attributed to the fact that expanding cultivated land to a certain extent can achieve economies of scale, facilitate the adoption of technologies, and enhance economic efficiency, thereby stimulating farmers' motivation to engage in grain production. Cooperative membership has a significantly positive impact on farmers' enthusiasm for grain cultivation. By integrating resources, empowering through technology, accessing markets, leveraging policy dividends, and extending industrial chains, cooperatives achieve cost reduction and efficiency improvement, thereby fundamentally enhancing farmers' enthusiasm for grain production. The impact of land transfer behavior on farmers' enthusiasm for grain cultivation is significantly positive. This may be attributed to the fact that land transfer facilitates land consolidation, enabling large-scale management. Such scaled operations effectively reduce production costs, enhance economic efficiency, and thereby stimulate farmers' motivation for grain cultivation. The proportion of agricultural income exhibits a significantly positive influence on farmers' enthusiasm for grain production. A plausible explanation is that a higher share of agricultural income implies more satisfactory returns from agricultural activities, incentivizing farmers to allocate greater efforts to agricultural production in pursuit of increased income, consequently elevating their enthusiasm for grain cultivation. Village economic development level demonstrates a significantly positive correlation with farmers' enthusiasm for grain production. This could be because villages with higher economic development levels provide timelier access to rice-related information, including market and policy updates, along with broader mechanization adoption, making farmers more inclined to cultivate double-cropping rice. Conversely, the distance from villages to county seats shows a significantly negative relationship with farmers' enthusiasm for grain cultivation. Potential reasons include the

**Table 3. Regression results of the impact of farmers' digital literacy on enthusiasm for grain cultivation.**

| VARIABLE | Double-cropping rice cultivation decision (Probit) | | Double-cropping rice cultivation area (Tobit) | |
|---|---|---|---|---|
| | (1) | (2) | (3) | (4) |
| Farmers' digital literacy | 0.947*** (0.187) | 0.719*** (0.252) | 1.864*** (0.251) | 0.923*** (0.261) |
| Gender | | 0.349*** (0.131) | | 0.530*** (0.141) |
| Age | | −0.008 (0.005) | | −0.017*** (0.006) |
| Educational attainment | | −0.022 (0.057) | | −0.098 (0.060) |
| Health status | | 0.037 (0.048) | | 0.063 (0.050) |
| Satisfaction with grain purchase prices | | 0.032 (0.053) | | −0.019 (0.055) |
| Number of agricultural laborers | | 0.033 (0.047) | | 0.063 (0.049) |
| Social capital | | −0.009 (0.018) | | −0.014 (0.019) |
| Cultivated land area | | 0.001* (0.000) | | 0.004*** (0.000) |
| Paddy field fertility | | 0.021 (0.054) | | 0.013 (0.057) |
| Cooperative membership | | 0.216* (0.112) | | 0.234** (0.115) |
| Land transfer behavior | | 0.300*** (0.091) | | 0.783*** (0.097) |
| Agricultural income share | | 0.006*** (0.001) | | 0.009*** (0.001) |
| Village economic development level | | 0.127** (0.061) | | 0.183*** (0.064) |
| Distance to county seat | | −0.004* (0.002) | | −0.004** (0.002) |
| N | 921 | 921 | 921 | 921 |
| Pseudo R2 | 0.0212 | 0.0944 | 0.0144 | 0.1304 |

*, **, and *** denote significance at the 10%, 5%, and 1% levels, respectively; values in parentheses are standard errors.

inconvenience of purchasing production inputs in remote areas and the likelihood of lower economic development levels in distant villages. These factors may lead local farmers to prioritize off-farm employment over double-cropping rice cultivation, resulting in diminished enthusiasm.

## 4.2 Endogeneity test

This study focuses on the impact of farmers' digital literacy on their enthusiasm for grain cultivation and its underlying mechanisms. However, endogeneity issues may arise in addressing this research question. First, as the micro-level survey data of farmers used in this paper only cover one year of information, it may fail to account for all potential influencing factors, leading to omitted variable bias. Second, reverse causality may exist, where farmers with higher enthusiasm for

grain cultivation might proactively seek professional knowledge in grain production, potentially resulting in higher digital literacy.

To mitigate endogeneity caused by omitted variables and reverse causality, this study employs the instrumental variable (IV) approach. Based on the research of scholar Li et al. and the reality of this study, we utilize the "municipal mobile phone penetration rate" as the IV for farmers' digital literacy [34]. First, the municipal mobile phone penetration rate satisfies the relevance condition. Mobile phones are the primary terminal for accessing digital information in rural China. A higher regional mobile phone penetration rate indicates better development of local mobile network infrastructure, which lowers the cost of access and use for residents. This creates an objective environment conducive to farmers' exposure to and engagement with digital tools (e.g., mobile apps, online payment, information search). Consequently, farmers in municipalities with higher penetration rates are more likely to develop digital skills, indicating a strong correlation between the IV and the endogenous variable (farmers' digital literacy). Second, the municipal mobile phone penetration rate plausibly satisfies the exclusion restriction. The penetration rate, as a macro-level indicator of infrastructure, does not directly determine an individual farmer's decision to plant double-cropping rice or their planting area. Its influence on cultivation behavior is primarily channeled through its effect on shaping the individual's digital literacy. We control for other potential confounding factors at the village level (e.g., economic development level, distance to the county seat) that might be correlated with both infrastructure and agricultural decisions. Furthermore, the variation in mobile phone penetration across municipalities is largely determined by historical investments in telecommunications infrastructure and regional topography, factors that can be considered exogenous to individual farmers' current grain production choices. Consequently, this instrument satisfies both the relevance and exogeneity conditions. The regression results using the IV approach are presented in Table 4.

First, regarding the dependent variable "double-cropping rice cultivation decision", the first-stage results show that the regression coefficient of the instrumental variable (IV) is significantly positive at the 5% level. This suggests that farmers in municipalities with higher mobile phone penetration rates exhibit higher levels of digital literacy. The F-statistic of 44.84 (>10) confirms that the IV satisfies the relevance requirement. The second-stage regression results demonstrate that digital literacy exerts a significantly positive impact on the double-cropping rice cultivation decision at the 1% level, indicating that enhanced digital literacy among farmers significantly promotes their adoption of double-cropping rice. This conclusion aligns with the baseline regression results. Additionally, the Cragg-Donald Wald F-statistic for weak instrument testing is 9.638, which exceeds the Stock-Yogo critical value of 8.96, suggesting no weak instrument issue.

Similarly, for the dependent variable "double-cropping rice cultivation area", the second-stage results reveal that digital literacy significantly affects the cultivation area at the 1% level, implying that improved digital literacy substantially increases farmers' cultivation area for double-cropping rice. This finding is consistent with the baseline regression results. The Cragg-Donald Wald F-statistic for weak instrument testing is 9.638, exceeding the Stock-Yogo critical value of 8.96 at the 15% level, further confirming the absence of a weak instrument problem.

**Table 4. Impact of farmers' digital literacy on enthusiasm for grain cultivation: instrumental variables.**

| VARIABLE | First stage: farmers' digital literacy | Second stage: double-cropping rice cultivation decision | Second stage: double-cropping rice cultivation area |
|---|---|---|---|
| Instrumental variable | 0.002*** (0.001) | 4.959*** (1.730) | 11.385*** (4.214) |
| Control variables | Controlled | Controlled | Controlled |
| Number of observations | 921 | 921 | 921 |
| F-statistic | 44.84 | | |
| Cragg-Donald Wald F | | 9.638 | 9.638 |

*, **, and *** denote significance at the 10%, 5%, and 1% levels, respectively; values in parentheses are standard errors.

## 4.3 Robustness tests

To further verify the reliability of the baseline regression results, this study conducts robustness checks by substituting regression models, altering the measurement method of the core explanatory variable, and applying winsorization.

(1) Model Substitution. Considering that the "double-cropping rice cultivation decision" is a binary variable and the "double-cropping rice cultivation area" is continuous, we replace the original model with Logit and OLS models, respectively. As shown in Columns (1) and (4) of Table 5, farmers' digital literacy remains significantly positively correlated with both the decision to cultivate double-cropping rice and the cultivation area, confirming the robustness of the baseline results.

(2) Alternative Measurement of Core Explanatory Variable. The core explanatory variable, farmers' digital literacy, has diverse measurement approaches in existing studies. To avoid bias from a single measurement method, we adopt the factor analysis approach proposed by Alakrash et al. to reconstruct the digital literacy index [35]. Results in Columns (2) and (5) of Table 5 indicate that farmers' digital literacy still enhances the scale of cultivation. This demonstrates that the baseline conclusions hold even when altering the measurement methodology, thereby validating the robustness of the findings.

(3) Winsorization. After applying 1% winsorization to the sample, the regression results (Columns (3) and (6) of Table 5) show that farmers' digital literacy significantly increases the likelihood of adopting double-cropping rice. The baseline conclusions remain robust, further supporting the reliability of the results.

## 4.4 Mechanism analysis

As suggested by theoretical analysis and research hypotheses, farmers' digital literacy may influence their enthusiasm for grain cultivation through three pathways: farmland transfer-in, agricultural machinery socialization services, and adoption of green grain production technologies. Given scholarly critiques of the traditional "three-step" mediation effect model, which may suffer from endogeneity issues, this study adopts the mediation test approach proposed by Bursztyn et al. for channel analysis [36]. Results are presented in Table 6.

(1) Farmland Transfer-in. Farmers' digital literacy exhibits a significantly positive effect on farmland transfer-in at the 1% statistical level, indicating that digital literacy facilitates farmland transfer-in and promotes the development of large-scale land management. Specifically, enhanced digital literacy enables farmers to access farmland transfer information more efficiently, increasing opportunities for land acquisition. After transferring in land, farmers can achieve scale

**Table 5. Robustness test results.**

| VARIABLE | Double-Cropping Rice Cultivation Decision | | | Double-Cropping Rice Cultivation Area | | |
|---|---|---|---|---|---|---|
| | Model (1) | Model (2) | Model (3) | Model (4) | Model (5) | Model (6) |
| Farmers' digital literacy | 1.170*** (0.418) | 0.274*** (0.097) | 0.696*** (0.254) | 0.923*** (0.264) | 0.223** (0.103) | 0.836*** (0.251) |
| Control variables | Controlled | Controlled | Controlled | Controlled | Controlled | Controlled |
| Pseudo R2 | 0.0952 | 0.0942 | 0.0964 | | 0.1283 | 0.1516 |
| R2 | | | | 0.4088 | | |
| N | 921 | 921 | 921 | 921 | 921 | 921 |

*, **, and *** denote significance at the 10%, 5%, and 1% levels, respectively; values in parentheses are standard errors. Model (1) replaces the baseline model with a Logit model; Model (4) replaces the baseline model with an OLS model; Models (2) and (5) use modified measurement methods for farmers' digital literacy under their respective baseline models.

**Table 6. Mechanism analysis of farmers' digital literacy on enthusiasm for grain cultivation.**

| VARIABLE | Farmland transfer-in | Agricultural machinery socialization services | Adoption of green grain production technologies |
|---|---|---|---|
| Farmers' digital literacy | 0.335*** (0.093) | 1.088*** (0.252) | 1.562*** (0.228) |
| Control variables | Controlled | Controlled | Controlled |
| Pseudo $R^2$ | 0.4119 | 0.0451 | 0.0460 |
| N | 921 | 921 | 921 |

*, **, and *** denote significance at the 10%, 5%, and 1% levels, respectively; values in parentheses are standard errors.

benefits by expanding cultivation areas, thereby improving production efficiency and cultivation motivation. Furthermore, improved digital literacy allows farmers to better evaluate land quality and market prospects, leading to more informed transfer-in decisions.

(2) Agricultural Machinery Socialization Services. Farmers' digital literacy shows a significantly positive impact on agricultural machinery socialization services at the 1% statistical level, suggesting that digital literacy enhances the utilization of such services. The adoption of agricultural socialization services reduces production costs and alleviates labor shortages. Farmers can optimize the use of machinery resources, minimize idle capacity, and improve operational efficiency through mechanization. This not only reduces labor intensity but also effectively substitutes manual labor, addressing workforce insufficiency.

(3) Adoption of Green Grain Production Technologies. Farmers' digital literacy significantly and positively influences the adoption of green grain production technologies at the 1% statistical level, demonstrating that digital literacy promotes the uptake of these technologies. Green production technologies optimize resource utilization, increase grain yields, reduce reliance on chemical pesticides and fertilizers, and enhance food safety and market competitiveness. Improved yield and quality directly boost farmers' economic returns, strengthening their cultivation motivation. Additionally, enhanced digital literacy enables farmers to better access and apply information on green production technologies, participate in relevant training programs, and ultimately improve technology adoption rates and implementation efficacy, further elevating cultivation enthusiasm.

## 4.5 Heterogeneity analysis

The empirical findings above robustly demonstrate that improving farmers' digital literacy significantly enhances their enthusiasm for grain cultivation. However, such effects may vary under different contextual conditions. This study further examines the multidimensional heterogeneous impacts of farmers' digital literacy on cultivation enthusiasm by analyzing subgroups based on educational attainment and the level of household part-time farming.

(1) Educational Attainment. Farmers with varying educational levels differ in their ability to acquire and process internet-based information. To investigate whether the effect of digital literacy on cultivation enthusiasm varies across educational backgrounds, we categorize farmers into low (primary school and below), medium (junior high school) and high (senior high school or above) educational attainment groups, following the methodology of Chen et al. [37]. Grouped regression results are presented in Table 7. Key findings include:

First, digital literacy in the low educational group shows no statistically significant effect on double-cropping rice cultivation decision and area, whereas it exerts a significantly positive impact in the high educational groups.

Second, digital literacy significantly increases double-cropping rice cultivation area in the medium and high educational groups, but the effect is more pronounced (higher significance level and larger regression coefficient) in the high educational group. This indicates heterogeneous impacts of digital literacy across educational levels.

**Table 7. Regression results by educational attainment groups.**

| VARIABLE | Low education group | | Medium education group | | High education group | |
|---|---|---|---|---|---|---|
| | Double-cropping rice cultivation decision | Double-cropping rice cultivation area | Double-cropping rice cultivation decision | Double-cropping rice cultivation area | Double-cropping rice cultivation decision | Double-cropping rice cultivation area |
| Farmers' digital literacy | 0.319 (0.491) | 0.608 (0.488) | 0.598 (0.372) | 0.780** (0.384) | 1.451*** (0.553) | 1.523*** (0.521) |
| Control variables | Controlled | Controlled | Controlled | Controlled | Controlled | Controlled |
| Pseudo R² | 0.0918 | 0.1223 | 0.1209 | 0.1409 | 0.1549 | 0.1651 |
| N | 336 | 336 | 394 | 394 | 191 | 191 |

*, **, and *** denote significance at the 10%, 5%, and 1% levels, respectively; values in parentheses are standard errors.

Mechanistically, farmers with higher education exhibit superior capabilities in acquiring digital information. They utilize internet resources to access double-cropping rice cultivation techniques, climate trends, and market dynamics, thereby improving farm management, mitigating risks, and enhancing production efficiency—all of which reinforce confidence in double-cropping cultivation. Additionally, better-educated farmers more effectively adopt digital tools such as smart agricultural machinery and IoT monitoring devices, enabling precision management and scaled production through expanded cultivation areas.

(2) The Level of Household Part-Time Farming. Household part-time farming is a typical feature of agricultural transformation in developing countries [38]. Its essence is the allocation of labor resources in the agricultural and non-agricultural sectors. As an important feature of farmers' economic behavior, the level of household part-time farming will affect the application of digital technology and grain production decisions. To examine the differential impacts of farmers' digital literacy on farmers' enthusiasm for grain cultivation across varying part-time farming levels, this study categorizes samples into four groups based on the proportion of agricultural income in total household income: pure agricultural households (≥95%), primary part-time farming households (50%−95%), secondary part-time farming households (5%−50%), and non-agricultural households (≤5%). Grouped regression results (Table 8) indicate that the promoting effect of farmers' digital literacy on their enthusiasm for grain cultivation increases significantly as part-time farming levels rise, as detailed below:

First, In the pure agricultural households group, the impact of farmers' digital literacy on their enthusiasm for grain cultivation is not statistically significant. Pure agricultural households take agriculture as the only source of income, and their planting decisions rely more on traditional experience and localized knowledge. The demand for digital technology focuses on basic production links, such as weather query and simple market information acquisition. However, the application of such technology has limited marginal benefit improvement for double-cropping rice cultivation, resulting in the insufficient promotion of digital literacy.

Second, the impact of digital literacy in the primary part-time farming households group is still not significant, but the direction of the coefficient is positive. Although it is mainly based on agriculture, it has begun to supplement its income through non-agricultural labor. The improvement of its digital literacy serves more non-agricultural fields, which has a certain crowding-out effect on the resource input of agricultural production, and weakens the direct pulling effect of digital literacy on the enthusiasm for grain cultivation.

Third, the impact of digital literacy in the secondary part-time farming households and non-agricultural households groups are significantly positive. The proportion of non-agricultural income in these groups is higher, and the improvement of their digital literacy is more inclined to feed back to agriculture through market-oriented mechanisms. On the one hand, with the help of digital platform to obtain accurate market information, optimize the planting structure and sales strategy; on the other hand, through online agricultural materials procurement, remote technical consultation, etc., production costs

**Table 8. Regression results by the level of household part-time farming subgroups.**

| VARIABLE | Pure agricultural households | | Primary part-time farming households | |
|---|---|---|---|---|
| | Double-cropping rice cultivation decision | Double-cropping rice cultivation area | Double-cropping rice cultivation decision | Double-cropping rice cultivation area |
| Farmers' digital literacy | −0.129 (0.909) | 0.885 (0.788) | 0.709 (0.688) | 0.501 (0.750) |
| Control variables | Controlled | Controlled | Controlled | Controlled |
| Pseudo R$^2$ | 0.2164 | 0.1811 | 0.1793 | 0.1394 |
| N | 122 | 122 | 159 | 159 |
| VARIABLE | Secondary part-time farming households | | Non-agricultural households | |
| | Double-cropping rice cultivation decision | Double-cropping rice cultivation area | Double-cropping rice cultivation decision | Double-cropping rice cultivation area |
| Farmers' digital literacy | 0.622* (0.374) | 0.941** (0.374) | 1.342** (0.555) | 0.733* (0.403) |
| Control variables | Controlled | Controlled | Controlled | Controlled |
| Pseudo R$^2$ | 0.0757 | 0.0905 | 0.0943 | 0.0467 |
| N | 406 | 406 | 234 | 234 |

*, **, and *** denote significance at the 10%, 5%, and 1% levels, respectively; values in parentheses are standard errors.

are reduced and grain yield expectations are improved. Especially in the non-agricultural households group, the influence coefficient of digital literacy on the decision-making of double cropping rice planting is 1.342 (5% significant), indicating that it has broken through the status of "part-timer" and is more inclined to use digital technology as a tool for agricultural management from the perspective of marketization, thus significantly stimulating the enthusiasm for grain cultivation.

## 5 Further discussion

This study integrates information search theory with farmer behavior theory to reveal the mechanisms through which digital literacy influences grain cultivation decisions. Unlike traditional research that focuses on single technology adoption (e.g., e-commerce platform usage), the multidimensional digital literacy index system (information acquisition, technology application, and data analysis) constructed in this paper more accurately reflects the enabling effects of digital technologies across the entire agricultural production chain. The findings indicate that digital literacy enhances farmers' risk perception [25] through expanded information channels (e.g., online policy interpretation, market trend tracking), leading farmers to perceive land transfer-in as a long-term investment rather than a short-term risk. The findings complement resource endowment theory, demonstrating that digital resources are emerging as an intangible and important production factor following land and labor [39], whose accumulation can amplify the productivity of traditional factors through the "digital leverage effect". Notably, the stronger effect of digital literacy among cooperative members validates the "organizational empowerment hypothesis". As information hubs and technology intermediaries, cooperatives reduce individual barriers to digital technology adoption through collective learning, providing theoretical support for nurturing new agricultural management entities. Additionally, the mediating effect of green production technology adoption suggests that digital literacy may serve as a critical variable in resolving the "environment-grain" trade-off, opening new avenues for research on micro-level incentive mechanisms in agricultural green transitions.

Although the study confirms the positive role of digital literacy, potential contradictions in policy implementation require vigilance: Deepening heterogeneity in the digital divide: Elderly farmers with lower education levels may face "technological displacement" due to digital skill deficiencies, necessitating safeguards against exacerbated group disparities during technology promotion. To avoid marginalizing elderly or low-education farmers, policies could introduce rural "digital envoys" to provide peer-based guidance in digital farming Structural imbalances in agricultural machinery service

markets: Current machinery cooperatives predominantly focus on plowing/harvesting stages, while high-end services such as drone-based pest control and intelligent monitoring remain insufficient, potentially limiting the realization of digital technology dividends. In response to this problem, high-end agricultural machinery equipment such as intelligent monitoring can be included in the scope of national agricultural machinery purchase subsidies, and the proportion of subsidies can be increased to directly reduce the procurement costs of farmers and cooperatives. Adaptability challenges in green technology promotion: As far as breeding technology is concerned, centralized breeding may not fully meet the needs of small farmers for varieties or variety groups that are more adaptable to the local environment, and participatory and decentralized methods and models need to be explored [40].

However, this study has certain limitations: Cross-sectional data constrain the exploration of dynamic relationships, regional sampling limits generalizability, instrumental variables may be confounded by local economic conditions, and subjective questionnaires entail potential reporting biases. Future research could incorporate panel data to track long-term effects, expand to other major grain-producing regions to validate conclusions, explore synergies between digital literacy and policies like ecological compensation, and integrate objective behavioral data to refine measurement methods. Beyond the stated limitations, two deeper considerations warrant attention: Phased characteristics of technology adoption: Digital technology application involves a three-stage progression--"tool utilization, process optimization, and model innovation". While this study emphasizes the initial tool utilization stage, future work should investigate the long-term impacts of technology internalization on production organization. Algorithmic interventions by digital platforms: Recommendation algorithms on e-commerce platforms may distort farmers' market information judgments. How such "digital manipulation" influences cultivation decisions merits further exploration.

In conclusion, this study provides a scientific foundation for digital technology-enabled agricultural transformation. Policymakers can harness "digital dividends" by improving infrastructure, strengthening skill training, and facilitating land transfers, thereby advancing national food security objectives.

## 6 Research conclusions and policy recommendations

### 6.1 Research conclusions

Based on the 2023 Jiangxi Province "Double Hundred and Double Thousand" farmer survey data, this study employs Probit and Tobit models to empirically analyze the impact of farmers' digital literacy on their enthusiasm for grain cultivation in traditional double-cropping rice production areas and explores the underlying mechanisms. The key findings are as follows:

First, improving farmers' digital literacy exerts a significantly positive influence on their enthusiasm for grain cultivation, promoting their adoption of double-cropping rice and expanding cultivation areas. This conclusion remains robust after substituting regression models, altering the measurement method of the core explanatory variable, and applying winsorization. Second, mechanism analysis reveals three pathways through which enhanced digital literacy affects cultivation enthusiasm: accelerating farmland transfer-in, facilitating the use of agricultural machinery socialization services, and promoting the adoption of green grain production technologies. And among the three mediating pathways, farmland transfer-in had the strongest effect, indicating that access to land remains a key channel for enhancing production behavior. Third, the impact of digital literacy on cultivation enthusiasm exhibits heterogeneity. Digital literacy demonstrates a more pronounced positive effect on double-cropping rice cultivation behaviors among farmers with higher educational attainment and the level of household part-time farming.

### 6.2 Policy recommendations

Based on the above research, the following policy recommendations are proposed to safeguard national food security, further enhance farmers' enthusiasm for grain cultivation, and incentivize double-cropping rice production in traditional core production areas:

1. Comprehensively accelerate the improvement of farmers' digital literacy. On the one hand, strengthen digital infrastructure by advancing rural informatization initiatives, including upgrading internet connectivity and data centers, to establish a robust hardware foundation for enhancing digital literacy. On the other hand, strengthen digital skills training for farmers. Fully utilize new media platforms to disseminate digital literacy content and improve farmers' ability to acquire and apply digital information. Simultaneously, incentivize research institutions, agricultural enterprises, and farmers' cooperatives to provide targeted digital technology training programs, enabling farmers to apply digital technologies (e.g., smart irrigation, drone monitoring) to agricultural production activities. A multi-channel, multi-stakeholder, and multi-method approach should be adopted to alleviate digital barriers and unlock farmers' "digital dividends".

2. Accelerate farmland transfer-in, expand agricultural operational scale, and promote agricultural machinery socialization services. On the one hand, encourage farmland transfer by improving rural land transfer platforms and mechanisms, expanding farmers' access to farmland information, enhancing transaction efficiency between supply and demand parties, and providing institutional support for farmland transfer-in and agricultural scale expansion, while creating conditions for the development of agricultural machinery socialization services. On the other hand, enhance the quality of agricultural machinery socialization services: First, increase government subsidies for agricultural machinery to ensure financial security. Second, actively cultivate specialized machinery service organizations, such as agricultural machinery cooperatives, to provide "one-stop" services spanning pre-production, production, and post-production stages (e.g., full-process mechanization, bulk input procurement, drying, and storage). Third, encourage new agricultural entities (e.g., technical associations, large-scale grain growers, professional service companies) to participate in machinery service provision, such as promoting advanced machinery to optimize equipment structures, enhance service accessibility, and boost farmers' cultivation enthusiasm.

3. Promote green grain production technologies to enhance cultivation enthusiasm. First, establish demonstration fields in suitable regions to showcase the advantages and practical outcomes of green production technologies, leveraging their exemplary role to stimulate farmers' interest in new technologies. Second, conduct extension seminars to strengthen farmers' technical awareness. Third, organize experts and technicians to provide on-site training in rural areas, including hands-on guidance, demonstrations, and technical consultations, to help farmers master the application methods and operational skills of new technologies. Fourth, increase financial subsidies for farmers adopting green production technologies to reduce production costs and risks. These multi-pronged measures will promote the adoption of green technologies, enhance cultivation motivation, and incentivize double-cropping rice production.

4. Implement differentiated digital empowerment strategies based on the level of household part-time farming. Accurately identify household types through a dynamic management database to enable targeted interventions. For pure agricultural households: Promote lightweight digital tools, provide "one-on-one" hands-on training via village service stations, and support micro-digital upgrades to enhance basic production efficiency. For primary part-time farming households: Deliver integrated "digital + agri-skills" training by embedding smart agriculture modules into vocational programs, and concurrently develop seasonal smart farming services to mitigate seasonal labor constraints. For secondary part-time farming households and non-agricultural households: Establish regional digital platforms integrating grain market prices and supply chain finance, and promote market-oriented models like cloud-based farm management and contract farming to strengthen data-driven cultivation decisions.

## Supporting information

**S1 Data. Data Sharing Declaration.**
(XLSX)

**S2 Data. Data-2025.**
(XLSX)

## Author contributions

**Conceptualization:** Zhipeng WANG.

**Data curation:** Yuqin Song.

**Formal analysis:** Yuqin Song, Zhihua Wu, Jianghua Chen.

**Investigation:** Huiping Xie, Chaoqun Li, Jianghua Chen.

**Methodology:** Yujie Hu.

**Project administration:** Chaoqun Li.

**Software:** Chaoqun Li, Yujie Hu.

**Supervision:** Yujie Hu, Jianghua Chen.

**Validation:** Huiping Xie, Zhihua Wu.

**Visualization:** Jianghua Chen.

**Writing – original draft:** Zhipeng WANG.

**Writing – review & editing:** Zhihua Wu.

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
