## [Decision Letter · Decision Letter 0]

23 Jul 2025

Dear Dr. WANG,

We look forward to receiving your revised manuscript.

Kind regards,

Dingde Xu

Academic Editor

PLOS ONE

Journal Requirements:

4. Your abstract cannot contain citations. Please only include citations in the body text of the manuscript, and ensure that they remain in ascending numerical order on first mention.

Reviewers' comments:

Reviewer's Responses to Questions

**Comments to the Author**

1. Is the manuscript technically sound, and do the data support the conclusions?

Reviewer #1: Yes

Reviewer #2: Yes

Reviewer #3: Yes

2. Has the statistical analysis been performed appropriately and rigorously?

Reviewer #1: Yes

Reviewer #2: Yes

Reviewer #3: N/A

3. Have the authors made all data underlying the findings in their manuscript fully available?

Reviewer #1: Yes

Reviewer #2: No

Reviewer #3: Yes

4. Is the manuscript presented in an intelligible fashion and written in standard English?

Reviewer #1: Yes

Reviewer #2: Yes

Reviewer #3: Yes

Reviewer #1: General Evaluation

Food security has always been a hot topic in academia and politics. Based on the survey data of farmers in Jiangxi Province, the author measures the enthusiasm of farmers for planting double cropping rice by their planting behavior, and empirically analyzes the impact and mechanism of farmers' digital literacy on the enthusiasm of farmers in traditional double cropping rice production areas in Jiangxi. In general, the topic selection has certain significance, the research design idea is clear, the demonstration process is reasonable, is a good empirical paper. In order to better improve the quality of the paper, several suggestions for reference:

Specific Suggestions for Revision

1.Title Optimization

The original title is overly lengthy. “Double Hundred and Thousand” refers to the survey project name rather than core concepts, which could be relocated to the data description section. Suggested revision: “Research on the Impact of Digital Literacy on Farmers’ Enthusiasm for Grain Cultivation: An Empirical Analysis Based on Double-Cropping Rice Cultivation in Jiangxi Province”.

2.Introduction Transition

The section from “the importance of food security” to “the role of digital literacy” lacks a logical bridge. Add transitional sentences to create a smoother connection between ideas.

3.Literature Gap Justification

The statement that “academic research on the impact of digital literacy on farmers’ enthusiasm for grain cultivation remains limited” needs to be substantiated by citing emerging studies (even those with conflicting conclusions) to precisely situate this paper’s contributions.

4.Introduction Streamlining

The introduction section could be appropriately condensed while maintaining clear logic and smooth transitions between ideas.

5.Mechanism Analysis Depth

The mechanism analysis could be more nuanced. For example, beyond information access, does digital literacy enhance farmers' market forecasting ability, risk tolerance, or operational management capacity when promoting farmland transfer? How does this differ from mere land transfer platform usage?

6.Model Equation Presentation

In the model explanation section, avoid inserting mathematical formulas in the main text to prevent inconsistent paragraph spacing.

7.Measurement Validity Concerns

Table 1 shows “digital technology literacy” measured by only one item (digital technology adoption). This may raise validity concerns and requires justification.

8.Instrumental Variable Validity

The “municipal mobile phone penetration rate” correlates strongly with regional economic development, which may directly affect grain cultivation enthusiasm through non-farm opportunities and infrastructure. Although “village economic development level” is controlled, potential omitted variable bias remains. The exogeneity of this instrumental variable and its limitations need more thorough discussion. Consideration of more micro-level instruments (e.g., village/community digital infrastructure) is advised.

9.Heterogeneity Analysis Robustness

In the education-level heterogeneity analysis, grouping “junior high school or below” vs “senior high school or above” is standard but requires reporting sample distribution (Table 7 shows N=730 vs N=191). The stability of results for the smaller sample group (senior high school or above) should be discussed. Similar sample size concerns apply to cooperative participation groups (non-members N=711 vs members N=210). Interaction term regression could enhance analysis.

10.Heterogeneity Dimension Expansion

The heterogeneity analysis could incorporate additional dimensions for more comprehensive exploration.

11.Policy Recommendation Precision

The policy suggestions section should better leverage findings from heterogeneity analysis to propose more targeted recommendations.

Reviewer #2: This manuscript presents a timely and policy-relevant investigation into how digital literacy influences grain cultivation enthusiasm among farmers in Jiangxi, China. By using data from a large-scale survey and employing Probity and Tobit models, the study provides robust empirical evidence to support its conclusions. The design of a multi-dimensional digital literacy index and the examination of mediation and heterogeneity pathways add methodological strength.

However, the manuscript would benefit from substantial improvements in clarity, depth of explanation, and structural precision, especially to meet international readership standards. The paper currently overuses general statements, lacks articulation of novel contributions, and could improve the interpretability of its empirical results.

Specific Comments and Suggestions for Revision

1. Clarify the Study’s Original Contributions

The current introduction does not clearly articulate the novelty of the paper. The authors are encouraged to list 2–3 explicit contributions at the end of the Introduction, e.g. The construction of a five-dimensional digital literacy index tailored to rural China; Identification of farmland transfer, machinery services, and green technology as mediating mechanisms; Heterogeneity analysis by education level and cooperative membership.

2. Define “Enthusiasm for Grain Cultivation” More Precisely

The term “enthusiasm” is used repeatedly but is operationalized via behavior (whether or not to grow double-cropping rice). Since "enthusiasm" typically refers to emotional or attitudinal variables, consider rephrasing or providing a clearer construct definition (e.g., “measured enthusiasm through behavioral proxies such as crop choice and cultivation area”).

3. Improve Figure 1 with Caption and English Terms

The theoretical framework figure lacks an informative caption and currently mixes Chinese-style conceptualization with English labeling. Revise the figure with: all variable names in English; A caption that explains arrows and variables (e.g., “This diagram illustrates the hypothesized mechanisms linking digital literacy to grain cultivation behavior.”)

4. Expand the Explanation of the Empirical Model

While the use of Probit and Tobit models is appropriate, the manuscript lacks clarity about variable definitions and the reasoning for model choice. Include a section that explicitly defines all variables and symbols in the model equations to enhance transparency.

5. Strengthen Justification of Instrumental Variable

The choice of “municipal mobile phone penetration rate” as an instrument for digital literacy needs stronger theoretical and empirical support. Authors should: Explain why this IV is correlated with digital literacy; Justify why it is not directly related to cultivation behavior; Cite prior literature or data on mobile infrastructure variation.

6. Standardize Tables and Improve Formatting

Capitalize variable names and avoid overly technical abbreviations; Report confidence intervals in addition to standard errors; Align table styles with PLOS ONE formatting standards (e.g., using bold for significant variables, consistent decimals).

7. Eliminate Redundancies and Vague Phrasing

Phrases such as “significantly positive effect on enthusiasm” are repeated frequently. Use more varied and precise language, e.g., “Digital literacy significantly increases the likelihood of adopting double-cropping rice” or “enhances the scale of cultivation.”

8. Summarize Mechanism Findings More Clearly in Conclusion

The current conclusion summarizes the baseline results well but underplays the mediation analysis. Add a brief sentence such as: “Among the three mediating pathways, farmland transfer-in had the strongest effect, indicating that access to land remains a key channel for enhancing production behavior.”

9. Prioritize Policy Recommendations

The policy section is comprehensive but could benefit from prioritization. For example, recommend first improving digital infrastructure, then farmer training, and finally green technology dissemination, according to feasibility and impact.

10. Deepen Discussion on Digital Divide

The “Further Discussion” section raises valuable points about digital inequality. Consider adding specific mitigation suggestions such as: “To avoid marginalizing elderly or low-education farmers, policies could introduce rural ‘digital envoys’ to provide peer-based guidance in digital farming.”

Reviewer #3: Thank you for the opportunity to read the paper titled Research on the Impact of Farmers' Digital Literacy on Farmers' Enthusiasm for Grain Cultivation: Based on the Survey Data of "Double Hundred and Thousand" Farmers in Jiangxi Province.

The paper makes a useful contribution to the literature but would benefit from revisions prior to publication. I offer some suggestions below.

The title needs revisions. The authors can consider the suggestion below

The Impact of Farmers' Digital Literacy on Farmers' Enthusiasm for Grain Cultivation: Findings from the "Double Hundred and Thousand" Farmers in Jiangxi Province

3 Data Source, Variable Setting and Model Selection

What was the study design? What impact design was used?

Results

The presentation of the results needs clarity.

Section 41. Baseline Regression Results. Why baseline results? How were you able to establish impact from only baseline study?

The presentation of the results should align with the study objectives/hypothesis.

6 Further Discussion

This section is out of place. The discussion of the findings should come before the conclusion.

**Do you want your identity to be public for this peer review?** For information about this choice, including consent withdrawal, please see our Privacy Policy

Reviewer #1: No

Reviewer #2: No

Reviewer #3: No

---

## [Author Response · Author response to Decision Letter 1]

11 Sep 2025

The Impact of Farmers’ Digital Literacy on Farmers’ Enthusiasm for Grain Cultivation: Findings from the “Double Hundred and Thousand” Farmers in Jiangxi Province

Response to Reviewers

We sincerely thank the editor and the reviewers for their precious time and invaluable comments, which were of great help in revising the manuscript. We have modified the manuscript accordingly and the detailed corrections are listed below point by point. We have used the same comment numbering that the reviewers used. The revised parts are shown in red in the resubmitted manuscript.

Response to Reviewer # 1

1.Title Optimization

The original title is overly lengthy. “Double Hundred and Thousand” refers to the survey project name rather than core concepts, which could be relocated to the data description section. Suggested revision: “Research on the Impact of Digital Literacy on Farmers’ Enthusiasm for Grain Cultivation: An Empirical Analysis Based on Double-Cropping Rice Cultivation in Jiangxi Province”.

Thank you for pointing out this! After reviewing the entire text, reconsider and revise the title of this article to “The Impact of Farmers’ Digital Literacy on Farmers’ Enthusiasm for Grain Cultivation: Findings from the ‘Double Hundred and Thousand’ Farmers in Jiangxi Province”.

2.Introduction Transition

The section from “the importance of food security” to “the role of digital literacy” lacks a logical bridge. Add transitional sentences to create a smoother connection between ideas.

Thank you for your valuable suggestion! We have added transitional sentences to the original text to ensure smooth transitions; the specific additions can be found in the red-marked sections of the manuscript.

3.Literature Gap Justification

The statement that “academic research on the impact of digital literacy on farmers’ enthusiasm for grain cultivation remains limited” needs to be substantiated by citing emerging studies (even those with conflicting conclusions) to precisely situate this paper’s contributions.

Thank you for pointing out this! The statement that “academic research on the impact of digital literacy on farmers’ enthusiasm for grain cultivation remains limited” lacks sufficient precision and rigor in its description; therefore, we propose revising that section to read: Moreover, the existing research has achieved certain results in the field of farmers’ grain production behavior and digital literacy effect, and it is of certain value to further explore the influence mechanism of farmers’ digital literacy on farmers’ enthusiasm for grain cultivation.

4.Introduction Streamlining

The introduction section could be appropriately condensed while maintaining clear logic and smooth transitions between ideas.

Thank you for your valuable suggestion! The introduction section has been appropriately condensed to ensure clear logic and natural transitions; the specific revisions can be found in the red-marked sections of the manuscript.

5.Mechanism Analysis Depth

The mechanism analysis could be more nuanced. For example, beyond information access, does digital literacy enhance farmers’ market forecasting ability, risk tolerance, or operational management capacity when promoting farmland transfer? How does this differ from mere land transfer platform usage?

Thank you for pointing out this! The text has explicitly stated that: “Farmers with high digital literacy typically leverage digital platforms to proactively access emerging trends in agriculture, rural areas, and farmer-related domains, breaking information barriers and translating external informational advantages into agricultural management capabilities and improving production efficiency.” The specific responses are presented as follows:

I. Core Role of Farmland Transfer-in and Empowerment Mechanism of Digital Literacy

Farmers’ digital literacy enhances enthusiasm for grain cultivation by accelerating farmland transfer-in, with the core logic being: digitally skilled farmers can more efficiently utilize digital tools to break information barriers, convert external information into agricultural management capabilities, thereby expanding operational scale and improving production efficiency. This mechanism can be analyzed through three dimensions: market prospect judgment capacity, risk resilience, and operational management capacity.

(I) Market Prospect Judgment Capacity: Transforming Information Advantages into Decision-Making Advantages

1.Information Acquisition and Price Forecasting

Leveraging digital platforms, farmers with high digital literacy can access real-time agricultural product prices, supply-demand data, and policy information, using data analysis to predict market trends. For instance, by integrating agrarian market data with meteorological information, farmers can accurately assess the profit expectations of double-cropping rice cultivation, avoiding “double-cropping to single-cropping” concealed farmland abandonment. A Jiangxi Province study also reveals that digitally skilled farmers excel in using online sales platforms to expand marketing channels, adjusting crop structures through price forecasting and directly curbing concealed farmland abandonment (Chen et al., 2025).

2.Policy Responsiveness and Opportunity Capture

Digital literacy enables farmers to swiftly respond to government policies such as “prime farmland for grain use” and agricultural subsidies. For example, accessing land transfer subsidy information via government platforms reduces farmland transfer-in costs and secures policy dividends.

(II) Risk Resilience: Digital Tools Mitigate Production and Market Risks

1.Production Risk Management

Disease Monitoring and Early Warning

Utilizing remote sensing data, meteorological information, and AI models, digitally skilled farmers can predict Fusarium head blight and rice blast risks 7 days in advance, reducing pesticide use and crop losses.

Digital Financial Support

Through privacy-computing platforms integrating land contract data with farmer credit information, digitally literate farmers gain easier access to low-interest loans, alleviating financial constraints during land transfer and enhancing risk resilience.

2.Market Risk Hedging

Diversified Sales Channels

Digital literacy enables farmers to engage in contract farming via e-commerce platforms, reducing reliance on traditional buyers. For example, blockchain technology for product traceability enhances brand premium capacity and stabilizes revenue expectations.

(III) Operational Management Capacity: Scaling and Efficiency Enhancement

1.Optimized Large-Scale Operation

Land Transfer Efficiency

Digitally skilled farmers use digital platforms to rapidly match land transfer-out and transfer-in parties, reducing information asymmetry. And digital finance reduces land transfer costs, facilitating operational scale expansion.

Resource Integration Capacity

Leveraging digital twin technology to create virtual farms enables real-time monitoring of equipment, crops, and markets. For instance, 3D modeling optimizes irrigation and fertilization plans, lowering per-unit production costs.

2.Technology Adoption and Innovation

Green Production Technology Application

Digitally literate farmers are more likely to adopt water-saving irrigation and smart agricultural machinery. Research shows digital literacy enhances double-cropping rice efficiency by promoting agricultural socialization services (Chen et al., 2025).

II. Conclusion

Farmers’ digital literacy accelerates farmland transfer-in and boosts grain cultivation enthusiasm through three pathways: accurate market prospect judgment, dynamic production risk control, and sustained operational efficiency optimization. This mechanism not only validates Hypothesis H2 but also reveals the multi-dimensional empowerment effects of digital literacy in agricultural modernization.

[1]Chen J H, Chen J, Qiu H L. The Impact of Digital Literacy on Implicit Farmland Abandonment ——A Study Based on Farmers’ “from Double to Single” Grain Cultivation. Journal of Huazhong Agricultural University (Social Sciences Edition).2025;01:15-30. https://10.13300/j.cnki.hnwkxb.2025.01.003

6.Model Equation Presentation

In the model explanation section, avoid inserting mathematical formulas in the main text to prevent inconsistent paragraph spacing.

Thank you for your valuable suggestion! The model selection section has had its formatting of formulas and symbols standardized; the specific revisions can be found in the red-marked sections of the manuscript.

7.Measurement Validity Concerns

Table 1 shows “digital technology literacy” measured by only one item (digital technology adoption). This may raise validity concerns and requires justification.

Thank you for pointing out this! The entropy method typically involves steps such as data normalization, calculating the proportion of each sample under each indicator, computing information entropy, and determining weights. Each indicator is processed independently, and theoretically, the weight of each indicator is determined based on its own level of variation, unrelated to the number of other indicators. Therefore, even if a certain dimension contains only one item, as long as the data for that item is processed correctly, its weight should be determined by the information capacity of the item itself, rather than the number of items in its group. Consequently, differences in the number of items across dimensions in the entropy method (e.g., “digital technology literacy” having only one item) generally do not directly affect the final calculation results.

8.Instrumental Variable Validity

The “municipal mobile phone penetration rate” correlates strongly with regional economic development, which may directly affect grain cultivation enthusiasm through non-farm opportunities and infrastructure. Although “village economic development level” is controlled, potential omitted variable bias remains. The exogeneity of this instrumental variable and its limitations need more thorough discussion. Consideration of more micro-level instruments (e.g., village/community digital infrastructure) is advised.

Thank you for your valuable suggestion! Next, we will thoroughly discuss the rationale behind the relevance and exogeneity of the instrumental variable “municipal mobile phone penetration rate”.

Theoretical Justification for the Instrumental Variable:

First, the municipal mobile phone penetration rate satisfies the relevance condition. Mobile phones are the primary terminal for accessing digital information in rural China. A higher regional mobile phone penetration rate indicates better development of local mobile network infrastructure, which lowers the cost of access and use for residents. This creates an objective environment conducive to farmers’ exposure to and engagement with digital tools (e.g., mobile apps, online payment, information search). Consequently, farmers in municipalities with higher penetration rates are more likely to develop digital skills, indicating a strong correlation between the IV and the endogenous variable (farmers’ digital literacy).

Second, the municipal mobile phone penetration rate plausibly satisfies the exclusion restriction. The penetration rate, as a macro-level indicator of infrastructure, does not directly determine an individual farmer’s decision to plant double-cropping rice or their planting area. Its influence on cultivation behavior is primarily channeled through its effect on shaping the individual’s digital literacy. We control for other potential confounding factors at the village level (e.g., economic development level, distance to the county seat) that might be correlated with both infrastructure and agricultural decisions. Furthermore, the variation in mobile phone penetration across municipalities is largely determined by historical investments in telecommunications infrastructure and regional topography, factors that can be considered exogenous to individual farmers’ current grain production choices.

The first-stage regression results (Table 4) confirm a statistically significant positive correlation between the IV and farmers’ digital literacy. The F-statistic exceeds the critical value of 10, robustly rejecting the hypothesis of a weak instrument. Therefore, the “municipal mobile phone penetration rate” is a valid and strong instrumental variable.

9.Heterogeneity Analysis Robustness

In the education-level heterogeneity analysis, grouping “junior high school or below” vs “senior high school or above” is standard but requires reporting sample distribution (Table 7 shows N=730 vs N=191). The stability of results for the smaller sample group (senior high school or above) should be discussed. Similar sample size concerns apply to cooperative participation groups (non-members N=711 vs members N=210). Interaction term regression could enhance analysis.

Thank you for pointing out this! The specific revisions are presented as follows:

For the heterogeneity analysis of educational attainment, we divided the sample into three groups: low educational attainment group, medium educational attainment group, and high educational attainment group, and conducted grouped regression. The grouped regression results are presented in the following table, with a relatively balanced distribution of sample sizes across the groups.

Regression results by educational attainment groups.

Variable Low education group Medium education group High education group

Double-cropping rice cultivation decision Double-cropping rice cultivation area Double-cropping rice cultivation decision Double-cropping rice cultivation area Double-cropping rice cultivation decision Double-cropping rice cultivation area

Farmers’ digital literacy 0.362

(0.481) 0.631

(0.518) 0.642*

(0.363) 0.831**

(0.414) 1.451***

(0.547) 1.523***

(0.515)

Control variables Controlled Controlled Controlled Controlled Controlled Controlled

Pseudo R2 0.0917 0.1207 0.1071 0.1339 0.1549 0.1651

N 336 336 394 394 191 191

Notes: *, **, and *** denote significance at the 10%, 5%, and 1% levels, respectively; values in parentheses are standard errors.

For the heterogeneity analysis of cooperative membership, due to the significant disparity in sample sizes between the two groups in the subgroup regression, an interaction term approach was initially adopted but yielded unsatisfactory results. Therefore, we revised the heterogeneity analysis framework to focus on the level of household part-time farming. Specifically, we categorized the sample into four groups: pure agricultural households, Primary part-time farming households, Secondary part-time farming households, and non-agricultural households, and conducted grouped regression. The results are presented in the following table, with a relatively balanced distribution of sample sizes across the groups.

Regression results by level of household part-time farming subgroups

Variable Pure agricultural households Primary part-time farming households

Double-cropping rice cultivation decision Double-cropping rice cultivation area Double-cropping rice cultivation decision Double-cropping rice cultivation area

Farmers’ digital literacy -0.165

(0.905) 0.814

(0.799) 0.737

(0.679) 0.577

(0.758)

Control variables Controlled Controlled Controlled Controlled

Pseudo R2 0.2143 0.1238 0.1602 0.1350

N 122 122 159 159

Variable Secondary part-time farming households Non-agricultural households

Double-cropping rice cultivation decision Double-cropping rice cultivation area Double-cropping rice cultivation decision Double-cropping rice cultivation area

Farmers’ digital literacy 0.627*

(0.372) 0.961**

(0.377) 1.352**

(0.555) 0.739*

(0.404)

Control variables Controlled Controlled Controlled Controlled

Pseudo R2 0.0712 0.0871 0.0905 0.0451

N 406 406 234 234

Notes: *, **, and *** denote significance at the 10%, 5%, and 1% levels, respectively; values in parentheses are standard errors.

10.Heterogeneity Dimension Expansion

The heterogeneity analysis could incorporate additional dimensions for more comprehensive exploration.

Thank you for your valuable suggestion! In the heterogeneity analysis section, we have incorporated an additional analysis of the level of household part-time farming. The specific resu

---

## [Decision Letter · Decision Letter 1]

28 Sep 2025

Research on the Impact of Farmers’Digital Literacy on Farmers’Enthusiasm for Grain Cultivation: Findings from the “Double Hundred and Thousand” Farmers in Jiangxi Province

PONE-D-25-33960R1

Dear Dr. WANG,

We’re pleased to inform you that your manuscript has been judged scientifically suitable for publication and will be formally accepted for publication once it meets all outstanding technical requirements.

Kind regards,

Dingde Xu

Academic Editor

PLOS ONE

Additional Editor Comments (optional):

The authors have addressed well with the reviewers' comments and it is suggested to be accepeted.

Reviewers' comments:

Reviewer's Responses to Questions

**Comments to the Author**

Reviewer #1: All comments have been addressed

Reviewer #2: All comments have been addressed

2. Is the manuscript technically sound, and do the data support the conclusions?

Reviewer #1: Yes

Reviewer #2: Yes

3. Has the statistical analysis been performed appropriately and rigorously?

Reviewer #1: Yes

Reviewer #2: Yes

4. Have the authors made all data underlying the findings in their manuscript fully available?

Reviewer #1: Yes

Reviewer #2: No

5. Is the manuscript presented in an intelligible fashion and written in standard English?

Reviewer #1: Yes

Reviewer #2: Yes

Reviewer #1: (No Response)

Reviewer #2: The author has made careful revisions and provided point-to-point responses to my questions. The manuscript is acceptable.

**Do you want your identity to be public for this peer review?** For information about this choice, including consent withdrawal, please see our Privacy Policy

Reviewer #1: No

Reviewer #2: No

---

## [Editor Report · Acceptance letter]

PONE-D-25-33960R1

PLOS One

Dear Dr. WANG,

I'm pleased to inform you that your manuscript has been deemed suitable for publication in PLOS One. Congratulations! Your manuscript is now being handed over to our production team.

Kind regards,

on behalf of

Dr. Dingde Xu

Academic Editor

PLOS One